# OPTIMIZING LATENT GOAL BY LEARNING FROM TRAJECTORY PREFERENCE

## ABSTRACT

A glowing body of work has emerged focusing on instruction-following policies for open-world agents, aiming to better align the agent's behavior with human intentions. However, the performance of these policies is highly susceptible to the initial prompt, which leads to extra efforts in selecting the best instructions. We propose a framework named *Preference Goal Tuning* (PGT). PGT allows an instruction-following policy to interact with the environment to collect several trajectories, which will be categorized into positive and negative samples based on preference. A preference optimization algorithm is used to fine-tune the initial goal latent representation using the collected trajectories while keeping the policy backbone frozen. The experiment result shows that with minimal data and training, PGT achieves an average relative improvement of $72.0\%$ and $81.6\%$ over 17 tasks in 2 different foundation policies respectively, and outperforms the best human-selected instructions. Moreover, PGT surpasses full fine-tuning in the out-of-distribution (OOD) task-execution environments by $13.4\%$, indicating that our approach retains strong generalization capabilities. Since our approach stores a single latent representation for each task independently, it can be viewed as an efficient method for continual learning, without the risk of catastrophic forgetting or task interference. In short, PGT enhances the performance of agents across nearly all tasks in the *Minecraft Skillforge* benchmark and demonstrates robustness to the execution environment.

## 1 INTRODUCTION

Recently, pre-training foundation policies in open-world environments with web-scale unlabeled datasets have become an increasingly popular trend in the domain of sequential control(Baker et al., 2022; Zhang et al., 2022; Collaboration et al., 2024; Brohan et al., 2023a; Yang et al., 2023). These foundation policies possess broad world knowledge, which can be transferred to downstream tasks. In the realm of foundation policies, there exists a category known as goal-conditioned policies, which are capable of processing input goals (instructions) and executing the corresponding tasks (Ding et al., 2019; Chane-Sane et al., 2021). The goal can be in different modalities, such as text instructions (Lifshitz et al., 2024), video demonstrations (Cai et al., 2023b), or multi-model instructions (Cai et al., 2024; Brohan et al., 2023b;a)).

However, much like large language models, these instruction-following policies are highly susceptible to the selection of "prompts"(Lifshitz et al., 2024; Wang et al., 2023b; Kim et al., 2024; Wang et al., 2023a). Researchers rely on trial and error to find the optimal prompt manually, and sometimes the quality of prompts doesn't align with human judgment. For instance, OpenVLA (Kim et al., 2024) shows a large performance gap when using "Pepsi can" compared to "Pepsi" as the prompt; for the same task of collecting wood logs, GROOT's performance varies significantly depending on the reference video used. Moreover, it is unclear whether an agent's failure to complete a task is due to the foundation policy's inherent limitations or the lack of a suitable prompt.

A common viewpoint from the LLM community thinks that most of the abilities are learned from the pre-training phase (Ouyang et al., 2022; Zhao et al., 2023a), while post-training is a method to elicit these abilities for solving tasks with rather small compute (Ziegler et al., 2020; Touvron et al., 2023; Lin et al., 2024). In this paper, we follow the roadmap of LLMs to consider post-training for the goal-conditioned foundation policies, hoping to improve downstream task performance efficiently

and effectively. On top of that, we identify several desiderata for the post-training for this type of policy:

- **Improved elicitation of pre-trained abilities.** This refers to (1) leveraging a broader range of abilities and (2) making it easier to harness these abilities, which leads to better performance on downstream tasks without the need for labor-intensive prompts.

- **Task environment generalization.** In open-world settings, a single task may be executed in vastly different contexts, making the policy's ability to generalize across environments crucial.

- **Efficient data exploitation.** As it's usually hard or expensive to collect training trajectory data for open-world foundation policy (Villalobos et al., 2024), the post-training is expected to be data-efficient. Meanwhile, it's also important to avoid over-fitting on the small amount of data.

- **Continued adaptation of tasks.** The ability to continually learn from experiences in open-world environments is crucial for generalist AI systems, and thus we expect the open-world foundation policy can continually learn more skills without degrading general ability.

To achieve these desiderata, we propose a framework named ***Preference Goal-Tuning*** (PGT). Firstly, an initial prompt is provided by humans, which may be suboptimal or not carefully refined. This task prompt is embedded into a *goal latent representation*, which is typically a high-dimensional vector. Next, PGT allows the foundation policy to interact with the environment under the guidance of the *goal latent representation*, for a small number of episodes ($\sim 10^2$ of trajectories in practice). These trajectories are then categorized into positive and negative samples based on designed rewards or human preferences. To elicit the ability from the pre-trained foundation policy, the backbone is fixed and a preference learning algorithm (Rafailov et al., 2024; Azar et al., 2024; Christiano et al., 2017; Hong et al., 2024) is applied to fine-tune the *goal latent representation* via collected trajectories. This training process can be iterative, as the fine-tuned *goal latent representation* can be used to recollect data once again.

We validate PGT in the open-ended Minecraft video game environment (Johnson et al., 2016), with 2 foundation policies and 17 tasks, in both in-distribution and out-of-distribution environments, showing that this framework can enhance performance for foundation policies across almost all tasks. For in-distribution settings, we achieved an average improvement of $72.0\%$ and $81.6\%$ in two different policies: GROOT (Cai et al., 2023b) and STEVE-1 (Lifshitz et al., 2024). For out-of-distribution settings, the figures are $73.8\%$ and $36.9\%$. We conduct extensive studies on different initial prompts and discover that PGT surpasses all human-selected prompts. Finally, we explore the potential of our method as an efficient approach to continual learning (CL). Since we only need to store a latent goal representation for each task in CL, our method is computationally light, storage-tight, with no fear of catastrophic forgetting or task interference in sight.

## 2 PRELIMINARY

### 2.1 SEQUENTIAL CONTROL

In sequential control settings, the environment is defined as a Markov Decision Process (MDP) $\langle \mathcal{S}, \mathcal{A}, \mathcal{R}, \mathcal{P}, d_0 \rangle$, where $\mathcal{S}$ is the state space, $\mathcal{A}$ is the action space, $\mathcal{R} : \mathcal{S} \times \mathcal{A} \to \mathbb{R}$ is the reward function, $\mathcal{P} : \mathcal{S} \times \mathcal{A} \to \mathcal{S}$ is the transition dynamics, and $d_0$ is the initial state distribution. A policy $\pi(a|s)$ interacts with the environment starting from $s_0 \sim d_9$. At each timestep $t \geq 0$, an action $a_t \sim \pi(a|s_t)$ is sampled and applied to the environment, after that, the environment transitions to $s_{t+1} \sim \mathcal{P}(s_t, a_t)$ and return reward $r_0 \sim \mathcal{R}(s_t, a_t)$. The goal of a policy is to maximize the expected cumulative reward $\mathbb{E}[\sum_{t=0}^{\infty} \gamma^t r_t]$, where $\gamma \in (0, 1]$ is a discount factor.

A goal-conditioned policy can be formulated as $\pi(a|s, g)$, where $g \in \mathcal{G}$ is a goal from goal space $\mathcal{G}$. The target of a goal-conditioned policy is to maximize the expected return $\mathbb{E}[\sum_{t=0}^{\infty} \gamma^t r_t^g]$, where $r_t^g$ is the goal-specific reward achieved at time step $t$.

## 2.2 GOAL-CONDITIONED POLICY

**GROOT**  GROOT (Cai et al., 2023b) is a goal-conditioned foundation policy trained on video data through self-supervised learning with a C-VAE(Sohn et al., 2015) framework. GROOT can follow video instructions in open-world environments. The instruction is encoded into a latent representation by the non-causal encoder, and the policy is a decoder module implemented by a causal transformer, which decodes the goal information in the latent space and translates it into a sequence of actions in the given environment states in an auto-regressive manner.

**STEVE-1**  STEVE-1 (Lifshitz et al., 2024) is also a goal-conditioned policy on Minecraft environment. STEVE-1 utilizes the goal latent representation of MineCLIP(Fan et al., 2022) to embed the future result video clip in dataset Andrychowicz et al. (2017), and fine-tunes a VPT model (Baker et al., 2022) as the policy network under the guidance of the MineCLIP embedding. As a C-VAE (Sohn et al., 2015) model is trained to predict "future video embedding" from text, STEVE-1 supports both text and video as instructions.

## 2.3 PREFERENCE LEARNING

While self-supervised learning models trained with large-scale parameters and data are experts in encoding knowledge, their outputs do not necessarily meet human intention. An effective solution is learning from preference-labeled data. Direct Preference Optimization(DPO) (Rafailov et al., 2024), as one method, serves as a way to directly optimize the model's outputs based on pair-wise positive-negative data. For a pair of responses $(y_1, y_2)$ corresponding to a prompt $x$, human labelers express their preference and classify them as win(w) and lose(l), denoted as $y_w \succ y_l \mid x$. Assuming we have a foundation model $\pi_{\text{ref}}$ and a dataset of preference $\mathcal{D} = \left\{ x^{(i)}, y_w^{(i)}, y_l^{(i)} \right\}_{i=1}^{N}$, DPO derives the optimization objective as:

$$\mathcal{L}_{\text{DPO}}(\pi_\theta; \pi_{\text{ref}}) = \mathbb{E}_{(x,y_w,y_l)\sim\mathcal{D}} \left[ -\log \sigma \left( \beta \log \frac{\pi_\theta(y_w \mid x)}{\pi_{\text{ref}}(y_w \mid x)} - \beta \log \frac{\pi_\theta(y_l \mid x)}{\pi_{\text{ref}}(y_l \mid x)} \right) \right]. \tag{1}$$

In addition to the DPO algorithm, IPO (Azar et al., 2024) proposed an improvement, enhancing the linearity of preference prediction and the fidelity to the reference model's outputs, KTO (Ethayarajh et al., 2024) optimized the model's output with the consideration of a human psychological effect *prospect theory* (Tversky & Kahneman, 1992). ORPO (Hong et al., 2024) further developed an empirical method for preference learning without a reference model. Previous to these works, SLiC (Zhao et al., 2022; 2023b) proposed calibration losses that also work empirically well and some of them are reference-model-free.

## 3 METHODOLOGY

### 3.1 PREFERENCE GOAL TUNING

In this section, we propose a novel policy post-training framework named ***Preference Goal-Tuning*** (PGT). This approach achieves significant performance improvements for foundation policies with minimal data and computational resources. Our method consists of two phases: the data collection phase and the training phase. An illustration of our method is in Figure 1. The details are as follows:

**Data Collection Phase**  We first select an initial prompt, which may be suboptimal or not carefully refined. This initial prompt is embedded into a high-dimensional vector by the encoder of the goal-conditioned policy. We allow the foundation policy to interact with the environment several times, collect $\sim 300$ synthetic trajectories, and divide them into positive trajectories and negative trajectories based on *human preference* or *reward from environment*.

When utilizing *human preference*, human annotators are required to label each trajectory as either positive (preferred) or negative (not preferred) based on their judgment. Since around 100 samples need to be annotated, the human labor cost remains manageable.

On the other hand, we utilize *reward from environment* for tasks like collect_wood(🟫), tool_bow(🏹) and explore_chest(📦). As reward information can be obtained from the

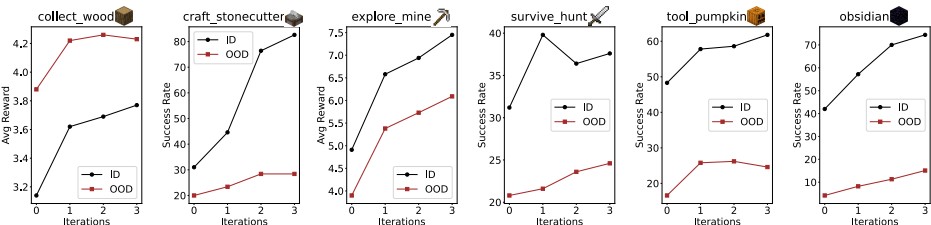

Figure 1: Pipeline of our Preference Goal Tuning (PGT). The process begins by selecting an initial prompt (can be video or text), encoding it into a latent representation, and allowing the policy to interact with the environment multiple times to collect trajectories. These trajectories are then classified as positive or negative based on human preferences or rewards. Then, the model is fine-tuned using the collected data, with only the latent goal embedding being trainable. Iterative training is supported.

Figure 2: Improvements with training iterations of our methods.

Minecraft simulator, we can directly use rewards as a supervisory signal for preference learning by selecting the top-performing trajectories as positive samples and the bottom-performing ones as negative samples for training.

**Training Phase** During the training phase, we adopted a learning approach to obtain an optimal *goal latent representation* - only the *goal latent representation* is trainable. Initially, we only leverage positive examples with traditional behavior cloning (BC) loss, but it does not yield the expected results. Recent studies have emphasized the importance of negative samples (Tajwar et al., 2024), prompting us to incorporate them into the training data. To reduce the agent's undesired behaviors and increase desired behaviors, the positive and negative samples are randomly combined into (win, lose) pairs for preference learning methods. Following the derivation approach of DPO, we obtained a loss for PGT in formula (2):

$$\mathcal{L}_{\text{PGT}}(g, g_{\text{ref}}) = -\mathbb{E}_{(\tau^{(w)}, \tau^{(l)}) \sim \mathcal{D}} \left[ \log \sigma \left( \beta \sum_{t=1}^{T} \log \frac{\pi(a_t^{(w)} \mid s_t^{(w)}, g)}{\pi(a_t^{(w)} \mid s_t^{(w)}, g_{\text{ref}})} - \log \frac{\pi(a_t^{(l)} \mid s_t^{(l)}, g)}{\pi(a_t^{(l)} \mid s_t^{(l)}, g_{\text{ref}})} \right) \right].$$
(2)

Details of derivation lies in Appendix A.1. Other preference learning algorithms such as SLiC (Zhao et al., 2022; 2023b) and IPO (Azar et al., 2024) are also feasible. Given the small amount of data and the limited number of trainable parameters, the training phase is relatively low-cost. Since the sample size is small, we use full gradient descent.

**Iterative Training** Our method supports iterative training. During the first training loop, the initial prompt is encoded into a *goal latent representation*, which we denote as $g_0$. According to 2, we set $g_{\text{ref}}$ as $g_0$ and initialize $g$ as $g_0$, then fine-tuning $g$ to $g_1$. We then use $g_1$ to recollect trajectories and repeat the training loop. Our experiments demonstrate that iterative training continues to improve performance for up to three rounds. See Figure 2 for iterative training details.

## 3.2 DESIGN CHOICES

In this section, we address the key design choices of our method and provide a comparative analysis of relevant baselines to justify why we use negative examples for preference learning and why we use parameter-efficient fine-tuning.

**Utilizing negative samples** A straightforward approach is to utilize only self-generated positive samples for behavior cloning (BC), and some studies have proved filtering and cloning is enough in many settings (Oh et al., 2018; Gulcehre et al., 2023). However, this approach does not explicitly indicate "which behaviors should be avoided", which is conducive to policy optimization (Tajwar et al., 2024). Incorporating negative data helps the policy distinguish between desirable and undesirable behaviors. As a comparison, we trained a version of the BC algorithm (with double data size of the positive samples to control the total amount of data) and conducted experiments with both soft prompt fine-tuning and full fine-tuning, and the results are listed in Table 1. We notice that when using BC algorithm, performance even declines in 3 out of 4 tasks in soft prompt fine-tuning.

Table 1: Performance improvements of the PGT-Loss over BC-Loss.

| Task | Soft Prompt | | | Full Fine-Tuning | | |
|---|---|---|---|---|---|---|
| | Pretained | BC-Loss | PGT-Loss | Pretained | BC-Loss | PGT-Loss |
| collect_wood | 3.14 | 3.28 | **3.62** | 3.14 | 3.26 | **3.46** |
| obsidian | 42.0 | 18.2 | **57.2** | 42.0 | 15.0 | **62.2** |
| explore_mine | 4.91 | 4.76 | **6.58** | 4.91 | 4.80 | **6.00** |
| tool_pumpkin | 48.3 | 45.4 | **57.8** | 48.3 | 48.6 | **58.4** |

**Tuning *goal latent representation* only** We compare the results of fine-tuning *goal latent representation* only and its counterpart that fine-tuning the entire policy model. There are two main reasons why we only fine-tune *goal latent representation*. First, fine-tuning the goal latent offers strong interpretability. For a goal-conditioned foundation policy trained through supervised learning with large datasets, the latent goal space usually holds abundant semantic meanings. However, since the human intention behind the instruction and the embedding in the goal space do not always align, the instructions selected by humans might not map well to the optimal latent representation in the goal space. Our method aims to obtain the optimal representation in goal space through a small amount of training. Second, due to the limited amount of data, full-parameter fine-tuning is highly prone to overfitting the training execution environment. For example, in Minecraft, task `collect wood`(🟫) requires the agent to collect logs from trees, regardless of the biome, seed, and initial location. With a small amount of training data, full-parameter fine-tuning tends to memorize environment-specific information to minimize the loss, which may result in reduced generalization ability.

The experimental results are consistent with our expectations. We find that in environments identical to the data collection phase (in-distribution environments, ID), soft-prompt tuning achieves comparable results to full fine-tuning. However, when rolling out in a different setting for the same task (out-of-distribution environments, OOD), the soft prompt method outperformes the full fine-tuning across all tasks. Detailed results are in Fig 3, and detailed numerical results are provided in Appendix C.2. The design of OOD settings is in Appendix B.4.

## 4 EXPERIMENTS

We select open-world Minecraft as the test bed to evaluate our methods (Lin et al., 2023; Fan et al., 2022). The tasks are selected from *Minecraft SkillForge* benchmark (Cai et al., 2023b). This benchmark covers over 30 diverse and representative tasks from 6 major categories. We put the details of this benchmark in Appendix B.2. Through our experiments, the following contributions of our method are verified:

- PGT remarkably improves the performance of two foundation policies, surpassing the best human-selected prompt.

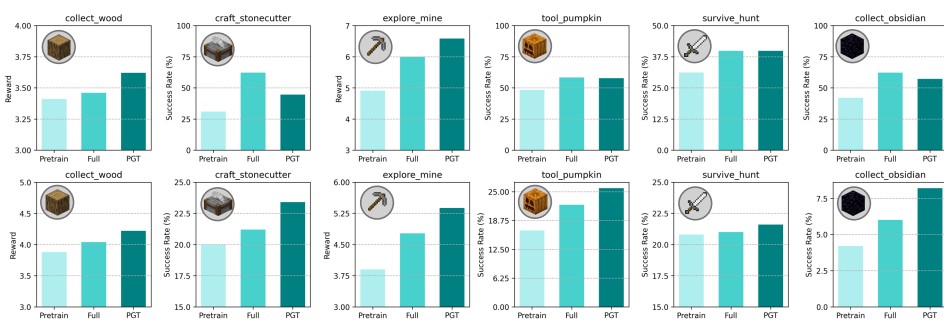

Figure 3: Comparison between full finetuning and PGT. Upper: In Distribution(ID). Lower: Out of Distribution(OOD).

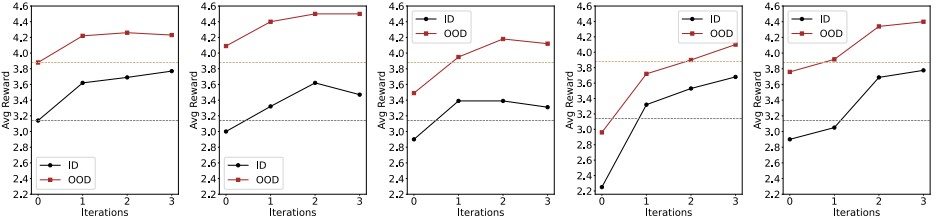

Figure 4: Different initial prompt results. Each line graph represents a different prompt, and the horizontal line represents the performance of the best human-selected prompt.

- PGT serves as an efficient continual learning method.

- PGT improves long-horizon task performance with a combination of planner and controller.

- PGT elicits skills that were not achievable with traditional prompts.

## 4.1 BOOSTING PERFORMANCE OVER PROMPT TUNING

Our approach significantly improves the instruction-following capability of the model. By fine-tuning specific aspects of the model's behavior, we achieve greater task performance compared to traditional prompt engineering techniques, which rely on manually crafted inputs. We discard tasks in *Minecraft SkillForge* that are too difficult (with zero success rate), or too easy (with a $100\%$ success rate and the specific value of the reward is meaningless).

We experimented with two foundation policies, GROOT and STEVE-1, in both in-distribution (ID) and out-of-distribution (OOD) settings. The modifications made to the OOD settings compared to the ID settings are detailed in Appendix B.4. For in-distribution settings, we achieved an average improvement of $72.0\%$ and $81.6\%$ in GROOT and STEVE-1 respectively. For out-of-distribution settings, the growths are $73.8\%$ and $36.9\%$. Results showed improvements for both models in both two settings across nearly all 17 tasks, with a particularly significant improvement in tasks like collect_dirt(⬛), craft_crafting_table(🟫), tool_flint (🗡). Detailed results can be found in Table 2.

**Different Initial Prompts** To validate the robustness of our method, we chose a representative task collect_wood(🟫), and selected 5 different initial prompts and performed iterative training on each. We found that, regardless of the initial prompt, the results after iterative training consistently outperformed the best human-selected reference video. This implies that for nearly any initial prompt, our method consistently surpasses even a carefully selected initial prompt by a human. We present the result in Figure 4.

Table 2: Success rates for different methods on tasks in *Minecraft SkillForge*. Δ represents the relative improvements of success rate between policy before and after post-training. GRO and STE represent the base policy of GROOT and STEVE-1 respectively. For tasks evaluated by success rate, the percentage sign (%) is omitted; the same applies to other parts of this paper.

| Task | In Distribution | | | | | | Out of Distribution | | | | | |
|---|---|---|---|---|---|---|---|---|---|---|---|---|
| | GRO | GRO+ | Δ | STE | STE+ | Δ | GRO | GRO+ | Δ | STE | STE+ | Δ |
| wood | 3.14 | **3.62** | 15.3% | 3.73 | **3.90** | 4.6% | 3.88 | **4.22** | 8.8% | 4.22 | **4.29** | 1.7% |
| dirt | 27.0 | **62.8** | 132.6% | 16.3 | **36.4** | 123.3% | 15.4 | **54.6** | 254.5% | 30.4 | **48.0** | 57.9% |
| wool | 30.4 | **40.8** | 34.2% | 43.3 | **56.6** | 30.7% | 34.0 | **41.6** | 22.4% | 45.6 | **60.2** | 32.0% |
| seagrass | 20.2 | **20.8** | 3.0% | 4.2 | **21.8** | 419.0% | 7.8 | **9.4** | 20.5% | 41.4 | **49.0** | 18.4% |
| stonecutter | 31.0 | **44.6** | 43.9% | 14.1 | **19.0** | 34.8% | 20.0 | **23.4** | 17.0% | 36.2 | **48.4** | 33.7% |
| ladder | 5.4 | **10.4** | 92.6% | 30.9 | **40.2** | 30.1% | 4.4 | **9.6** | 118.2% | 29.6 | **41.2** | 39.2% |
| enchant | 15.0 | **18.4** | 22.7% | 0 | 0 | - | 19.4 | **21.8** | 12.4% | 0 | 0 | - |
| crafting_table | 5.4 | **14.6** | 170.4% | 4.0 | **9.6** | 140.0% | 6.0 | **18.4** | 206.7% | 2.0 | **6.4** | 220.0% |
| mine | 4.91 | **6.58** | 34.0% | 6.46 | **7.32** | 13.3% | 3.9 | **5.38** | 37.9% | 3.49 | **5.37** | 53.9% |
| chest | 15.7 | **21.2** | 35.0% | 3.4 | **4.2** | 23.5% | **38.4** | 38.2 | -0.5% | 0.5 | **0.6** | 20.0% |
| hunt | 31.2 | **39.8** | 27.6% | **2.9** | 1.0 | -65.5% | 20.8 | **21.6** | 3.8% | **1** | 0.2 | -80.0% |
| combat | 31.7 | **36.6** | 15.5% | 0 | 0 | - | 83.4 | **85.6** | 2.6% | 0 | 0 | - |
| plant | 2.71 | **3.09** | 14.0% | 1.74 | **1.81** | 4.0% | 2.85 | **3.11** | 9.1% | 1.79 | **1.94** | 8.4% |
| pumpkin | 48.3 | **57.8** | 19.7% | 1.3 | **6.2** | 376.9% | 16.6 | **25.8** | 55.4% | 7.6 | **14.0** | 84.2% |
| bow | 77.4 | **85.8** | 10.9% | 88.9 | **97.8** | 10.0% | 77.4 | **90.6** | 17.1% | 65.2 | **88.0** | 35.0% |
| flint | 1.2 | **7.4** | 516.7% | 73.6 | **76.6** | 4.1% | 1.2 | **5.8** | 383.3% | 48.0 | **52.0** | 8.3% |
| obsidian | 42.0 | **57.2** | 36.2% | 0.4 | **0.7** | 75.0% | 4.2 | **8.2** | 95.2% | 0 | 0 | - |

## 4.2 EFFICIENT CONTINUAL LEARNING

Our method is an efficient approach to continual learning, as it requires only minimal training for each task, followed by storing a high-dimensional latent (typically consisting of a few hundred floating-point values) as a task representation. As a result, our method avoids issues like catastrophic forgetting and task interference.

We compare PGT with multiple continual learning baselines: multi-task learning (MTL), naive continual learning(NCL), knowledge distillation (KD) (Hinton et al., 2015), experience replay (ER) (Lopez-Paz & Ranzato, 2022), elastic weight consolidation (EWC) (Kirkpatrick et al., 2017). It's worth mentioning that every continual learning baseline is conducted under full fine-tuning, which has a parameter size several orders of magnitude times larger than ours. We first implemented the multi-task learning (MTL) baselines on six representative tasks, with the results presented in Table 3. We find that, similar to the results of full-parameter fine-tuning, our method achieved comparable performance to MTL in ID settings, while surpassing MTL in OOD settings.

We experiment in the following order: collect_obsidian(⬛) → tool_pumpkin(🟧) → craft_crafting_table(🟫) → explore_climb(🧗). The result after continual learning 4 tasks is in Table 4, and we place the detailed result of continual learning after each task in Appendix C.4. We conduct experiments of naive continual learning (NCL) (Table 10), knowledge distillation (KD)(Table 11), experience replay (ER)(Table 12), and elastic weight consolidation (EWC) (Table 13).

Experiment results show that in addition to being more efficient in terms of computational resources and storage, our method excels in handling diverse tasks, demonstrating superior generalization capabilities. In out-of-distribution settings, we outperform the ensemble in each of the 6 tracks, and we achieve comparable results to MTL.

## 4.3 SOLVING LONG-HORIZON CHALLENGES WITH PLANNER

It is a common approach to combine a high-level planner and a low-level controller for functionality and versatility. We combine the GROOT agent with JARVIS-1 planner (Wang et al., 2023b), trying to craft items from scratch spawning in a forest with random initial orientation and angle. JARVIS-1 also offers an API script for crafting items. We give the agent 1000 timesteps to run and select five representative items in the wood-related tech tree. We observe improvements in long-horizon task

Table 3: Multitask learning on Minecraft different tasks.

| Task | In Distribution(ID) | | | | Out of Distribution(OOD) | | | |
|---|---|---|---|---|---|---|---|---|
| | pretrained | ensemble | MTL | Ours | pretrained | ensemble | MTL | Ours |
| collect_wood | 3.14 | 3.46 | **3.64** | 3.62 | 3.88 | 4.04 | **4.30** | 4.22 |
| craft_stonecutter | 31.0 | 62.2 | **66.8** | 44.6 | 20.0 | 21.2 | 18.6 | **23.4** |
| explore_mine | 4.91 | 6.00 | 5.98 | **6.58** | 3.90 | 4.77 | 4.70 | **5.38** |
| survive_hunt | 31.2 | 39.8 | **44.2** | 39.8 | 20.8 | 21.0 | **31.4** | 21.6 |
| tool_pumpkin | 48.3 | 58.4 | **61.4** | 57.8 | 16.6 | 22.2 | 22.8 | **25.8** |
| collect_obsidian | 42.0 | **62.2** | 53.2 | 57.2 | 4.2 | 6.0 | **10.2** | 8.2 |

Table 4: Different continue learning baselines.

| Task | In Distribution(ID) | | | | | Out of Distribution(OOD) | | | | |
|---|---|---|---|---|---|---|---|---|---|---|
| | ER | EWC | KD | NCL | PGT | ER | EWC | KD | NCL | PGT |
| collect_obsidian | 60.2 | 64.6 | **66.8** | 61.2 | 57.2 | 6.0 | 5.4 | 5.4 | 6.8 | **8.2** |
| tool_pumpkin | **65.4** | 60.0 | 60.8 | 61.4 | 57.8 | 25.0 | 23.8 | 20.6 | 20.4 | **25.8** |
| craft_table | 8.6 | 6.8 | 6.8 | 7.2 | **14.6** | 9.0 | 7.4 | 5.8 | 7.0 | **18.4** |

performance compared to the baseline, which is shown in Table 5. This finding demonstrates the soft prompts trained with PGT have strong robustness and environmental generalization, and have the potential to serve as a bridge between the planner and the controller in the policy post-training stage.

## 4.4 ELICITING NEW SKILLS

For task tool_trident(⚒), given standard gameplay videos, the agent was unable to complete the task. As a result, the standard PGT pipeline cannot collect positive data. Instead, we recorded 20 trajectories by humans and trained with behavior cloning. Even though the success rate was still low, we found several success examples, meaning that the agent acquired the ability to complete the task. This implies that during the pretraining phase, the agent already possessed the ability to complete the task, but lacked the appropriate prompt to elicit this ability. Our method, through minimal training on the soft prompt, successfully activated this capability.

## 4.5 ABLATION STUDY ON PEFT METHODS

We compare our method with other parameter-efficient fine-tuning (PEFT) methods: LoRA (Hu et al., 2021), BitFit (Zaken et al., 2022) and VeRA (Kopiczko et al., 2024). We still utilize P-N samples for PGT for all of them fine-tuning the entire model. We found that our method performed well among the four methods. Moreover, in task expore_mine(⚒) and collect_obsidian(⬢), LoRA fine-tuning also demonstrated promising results. The result is in Figure 5, and the numerical result is in Appendix C.3.

Table 5: Long Horizon Task: Craft object from scratch. The numbers represent success rate (%)

| Task | Wooden Stick | Wooden Sword | Oak Boat | Oak Wood | Large Chest |
|---|---|---|---|---|---|
| Pretrain | 99.5 | 94.0 | 80.7 | 60.8 | 37.8 |
| PGT | **100** | **100** | **89.5** | **80.7** | **64.9** |

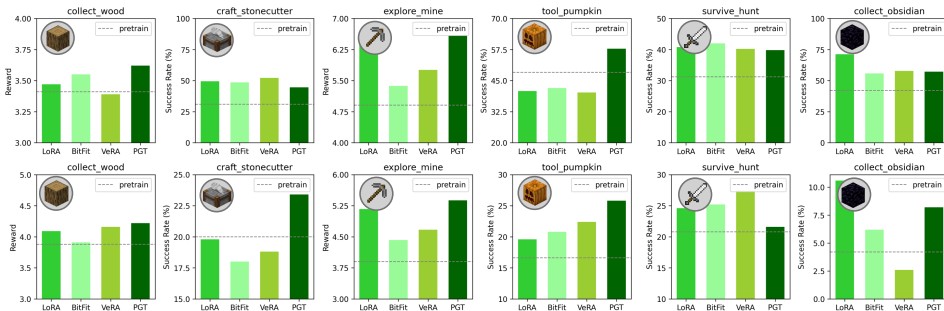

Figure 5: Result of different parameter efficient methods. The horizontal line indicates pretraining performance. Upper: ID. Lower: OOD.

## 5 RELATED WORK

### 5.1 FOUNDATION MODELS FOR DECISION-MAKING

Foundation models have gained huge success in the field of language (Brown et al., 2020; OpenAI, 2024) and vision (He et al., 2016; Kirillov et al., 2023), and an increasing number of studies are exploring the potential of foundation models in sequential control (Yang et al., 2023; Zhang et al., 2023; Wang et al., 2023a; Cai et al., 2023a; Cheng et al., 2024). VPT (Baker et al., 2022) is a foundation policy pretrained by video data behavior cloning and fine-tuned by reinforcement learning, which is capable of obtaining diamonds from scratch in Minecraft. Lifshitz et al. (2024) adapted the VPT model to following human instructions under the guidance of MineCLIP (Fan et al., 2022) and Cai et al. (2023b) started from scratch to train a Minecraft instruction-following agent controlled by the CVAE posterior distribution, which solves a variety of tasks in the open-world environment. In the field of robotics, there are also many foundation policies like BC-Z (Jang et al., 2022), GATO (Reed et al., 2022), RT-1 (Brohan et al., 2023b), RT-2 (Brohan et al., 2023a) and VQ-BeT (Lee et al., 2024).

### 5.2 PREFERENCE LEARNING

Directly obtaining high-quality human annotations, such as expert numerical ratings (Akrour et al., 2014; Fürnkranz et al., 2012), or expert demonstrations (Silver et al., 2016), is often extremely time-consuming, labor-intensive, and brain-consuming to annotators (Knox & Stone, 2009). Fortunately, the cost is greatly reduced by letting them label pairs or groups of data with simply their preferences Christiano et al. (2017). As a fruitful method to leverage more low-annotation-difficulty data, preference learning has been studied extensively in recent years. Christiano et al. (2017); Ziegler et al. (2020); Ouyang et al. (2022) utilized preference data to teach a reward model, and conducted reinforcement learning on sequential decision-making games or language modeling, demonstrating the efficiency and wide application of preference learning. These methods rely on another model for simulating the reward function and on-policy data. Therefore, some simpler alternatives that do not require reinforcement learning soon emerged (Rafailov et al., 2024; Azar et al., 2024; Meng et al., 2024) or even without reference model for regularization (Hong et al., 2024). Even though these methods do not strictly demand on-policy data, researchers (Tajwar et al., 2024) found that preference pairs generated by the current policy can improve fine-tuning efficiency.

## 6 LIMITATIONS AND FUTURE WORK

PGT has shown remarkable capability in improving task performance. However, it still has some limitations and untapped potential awaiting further exploration.

**Limitations** PGT requires multiple interactions with the environment to obtain positive and negative samples. While this is feasible in simulated environments like Minecraft, in other domains, such as robotics, the cost of interacting with the environment can be very high, or opportunities for

interaction may be limited (due to the risk of damage to the robots). In such cases, PGT may not be suitable.

**Potentials**  Our method holds significant potential. First, all of our experiments were conducted in the Minecraft environment, but there are many instruction-following policies in the robotics domain as well. We believe that PGT could also achieve promising results in robotics. Second, the current experiments only cover several simple long-horizon tasks, like building a large chest from scratch. We are thrilled to explore how PGT can help solve longer and more complex tasks in Minecraft, like the ultimate goal: killing the ender dragon.

## 7 CONCLUSION

We have introduced a framework named *Preference Goal-Tuning* (PGT), which is an efficient post-training method for foundation policies. It utilizes a small amount of human preference data to fine-tune *goal latent* in goal-conditioned policies. PGT significantly enhances the capability of the foundation policy with minimal data and training, easily surpassing the best human-selected instructions. Our method also demonstrates the potential for acquiring new skills and serving as an efficient method for continual learning.

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

# A  MATHEMATICAL DERIVATION

## A.1  PGT LOSS

Our PGT method is based on preference learning with a sequential decision-making process. Our policy is formulated as $\pi(\tau|g)$, meaning the probability of generating trajectory $\tau$ under latent goal $g$. Assume $\tau = (s_t, a_t)_{t=0}^{N-1}$ is a $N$ step trajectory , $\pi(\tau|g)$ can be expanded as:

$$\pi(\tau|g) = \prod_{i=0}^{N-1} \pi(a_i|s_i, g)p(s_{i+1}|s_t, a_t) \tag{3}$$

Generally, we want to utilize human preference to finetune our policy. Take DPO as an example, "preference" is assumed to be generated by an oracle reward function $r^*(\tau)$, which is inaccessible. $r^*(\tau)$ represents how well trajectory $\tau$ performs the task. The better $y$ performs, the higher $r^*(\tau)$ is. Even though we cannot obtain this oracle reward in practice, we can still set it as our objective:

$$\max_g \mathbb{E}_{\tau \sim \pi(\tau|g)}\big[r^*(\tau)\big] - \beta \mathbb{D}_{\text{KL}}\big[\pi(\tau|g) \,||\, \pi(\tau|g_{\text{ref}})\big] \tag{4}$$

Here $g$ is the *latent goal*, which is trainable, and $g_{ref}$ is the initial goal latent. The first term is to maximize the reward, and the second term is to constrain the trained $g$ such that it does not deviate too far from $g_{ref}$. By applying the same derivation method as DPO, we have:

$$\max_g \mathbb{E}_{\tau \sim \pi(\tau|g)}\big[r^*(\tau)\big] - \beta \mathbb{D}_{\text{KL}}\big[\pi(\tau|g) \,||\, \pi(\tau|g_{\text{ref}})\big] \tag{5}$$

$$= \max_g \mathbb{E}_{\tau \sim \pi(\tau|g)}\Big[r^*(\tau) - \beta \log \frac{\pi(\tau; g)}{\pi(\tau; g_{\text{ref}})}\Big] \tag{6}$$

$$= \max_g \mathbb{E}_{\tau \sim \pi(\tau|g)}\Big[\frac{r^*(\tau)}{\beta} - \log \frac{\pi(\tau|g)}{\pi(\tau|g_{\text{ref}})}\Big] \tag{7}$$

$$= \min_g \mathbb{E}_{\tau \sim \pi(\tau|g)}\Big[-\frac{r^*(\tau)}{\beta} + \log \frac{\pi(\tau|g)}{\pi(\tau|g_{\text{ref}})}\Big] \tag{8}$$

$$= \min_g \mathbb{E}_{\tau \sim \pi(\tau|g)}\Big[\log \frac{\pi(\tau|g)}{\exp(\frac{r^*(\tau)}{\beta})\pi(\tau|g_{\text{ref}})}\Big] \tag{9}$$

$$= \min_g \mathbb{E}_{\tau \sim \pi(\tau|g)}\Big[\log \frac{\pi(\tau|g)}{\frac{1}{Z}\exp(\frac{r^*(\tau)}{\beta})\pi(\tau|g_{\text{ref}})} - \log Z\Big] \tag{10}$$

$$\tag{11}$$

where $Z = \sum_\tau \pi(\tau|g_{\text{ref}}) \exp\left(\frac{r^*(\tau)}{\beta}\right)$. We define $g^*$ that satisfied

$$\pi(\tau|g^*) = \frac{\exp(\frac{r^*(\tau)}{\beta})\pi(\tau|g_{\text{ref}})}{Z} \tag{12}$$

The training object becomes:

$$\min_g \mathbb{E}_{\tau \sim \pi(\tau|g)}\left[\log \frac{\pi(\tau|g)}{\frac{1}{Z}\exp(\frac{r^*(\tau)}{\beta})\pi(\tau|g_{\text{ref}})} - \log Z\right] \tag{13}$$

$$= \min_g \mathbb{E}_{\tau \sim \pi(\tau|g)}\left[\log \frac{\pi(\tau|g)}{\pi(\tau|g^*)} - \log Z\right] \tag{14}$$

$$= \min_g D_{KL}(\pi(\tau|g)||\pi(\tau|g^*)) - \log Z \tag{15}$$

So we can obtain closed-form optimal solution:

$$\pi(\tau|g) = \pi(\tau|g^*) = \frac{\exp(\frac{r^*(\tau)}{\beta})\pi(\tau|g_{\text{ref}})}{Z} \tag{16}$$

$$q \tag{17}$$

Consider the Bradly-Terry(BT) model:

$$p(\tau_1 \succ \tau_2) = \frac{\exp(r(\tau_1))}{\exp(r(\tau_1)) + \exp(r(\tau_2))}. \tag{18}$$

fill Eq. 17 into Eq. 18, we have:

$$p(\tau_1 \succ \tau_2) = \sigma\left(\beta \log \frac{\pi(\tau_1|g^*)}{\pi(\tau_1|g_{\text{ref}})} - \beta \log \frac{\pi(\tau_2|g^*)}{\pi(\tau_2|g_{\text{ref}})}\right). \tag{19}$$

Decompose $\tau$ into factors, filling in equation 3, we can get:

$$\log p(\tau^{(w)} \succ \tau^{(l)}) \tag{20}$$

$$= \log \sigma\left(\beta \sum_{t=1}^T \log \frac{\pi(a_t^{(w)} \mid s_t^{(w)}, g^*)}{\pi(a_t^{(w)} \mid s_t^{(w)}, g_{\text{ref}})} - \log \frac{\pi(a_t^{(l)} \mid s_t^{(l)}, g^*)}{\pi(a_t^{(l)} \mid s_t^{(l)}, g_{\text{ref}})}\right). \tag{21}$$

Finally, our optimization objective becomes:

$$\mathcal{L}_{\text{PGT}}(g, g_{\text{ref}}) = -\mathbb{E}_{(\tau^{(w)}, \tau^{(l)}) \sim \mathcal{D}}\left[\log \sigma\left(\beta \sum_{t=1}^T \log \frac{\pi(a_t^{(w)} \mid s_t^{(w)}, g)}{\pi(a_t^{(w)} \mid s_t^{(w)}, g_{\text{ref}})} - \log \frac{\pi(a_t^{(l)} \mid s_t^{(l)}, g)}{\pi(a_t^{(l)} \mid s_t^{(l)}, g_{\text{ref}})}\right)\right]. \tag{22}$$

## B  EXPERIMENT DETAILS

### B.1  MINECRAFT

Minecraft is a popular sandbox game that allows players to freely create and explore their world. Since Minecraft is an open-world environment, many recent works have designed agents and conducted explorations within Minecraft (Johnson et al., 2016). In this work, we conduct experiments on 1.16.5 version MineRL (Guss et al., 2019) and MCP-Reborn.

### B.2  MINECRAFT SKILLFORGE BENCHMARK

Minecraft SkillForge Benchmark is a comprehensive task suite that covers various types of tasks in Minecraft. All tasks are categorized into six major groups:

- Collect task: these tasks are designed to evaluate an AI agent's capability in resource acquisition proficiency and spatial awareness.

- Craft task: these tasks are designed to shed light on an AI agent's prowess in item utilization, the intricacies of Minecraft crafting mechanics, and the nuances of various game mechanic interactions.

- Explore task: these tasks are designed to evaluate an AI agent's navigation proficiency, understanding of diverse environments, and intrinsic motivation for exploration.

- Survive task: these tasks are designed to analyze an AI agent's ability to ensure its survival, adeptness in combat scenarios, and capability to interact with the environment to meet basic needs.

- Tool task: these tasks are designed to deeply investigate an AI agent's capabilities in tool utilization, precision in tool handling, and contextual application of various tools to carry out specific tasks.

- Build task: these tasks are devised to evaluate an AI agent's aptitude in structural reasoning, spatial organization, and its capability to interact with and manipulate the environment to create specific structures or outcomes.

## B.3 TASK METRICS AND SELECTION

For most tasks, the environment logs the rewards when the corresponding objectives are achieved. We define tasks with a reward function greater than 0 as successful, and the frequency of successfully completing a task is referred to as the success rate. However, tasks like "collect_wood" "explore_mine" and "survive_plant" have a success rate of over 95% across different agents, and the specific values of the reward function are meaningful, reflecting the agents' capabilities in these tasks, so we use the detailed reward value as the metric.

We removed the tasks that are too easy that agents can perform a success rate of 100% while the specific value of the reward is either high enough (e.g. collect_grass) or not meaningful (e.g. survive_sleep). Also, to simplify the experiment, We removed the tasks for which the reward function cannot be directly obtained from the game, including subjective tasks (e.g. building tasks) and objective tasks where the environment does not log explicit rewards (e.g. craft_smelt). Moreover, mining obsidian is a high requirement for the agent's sensitivity to the objectives, and the agent needs to stay focused on the same goal over extended time steps to perform useful actions; therefore, we consider this task to be quite important and add it to the testing tasks apart from *Minecraft SkillForge*.

## B.4 OUT-OF-DISTRIBUTION SETTINGS

We designed the out-of-distribution (OOD) setting with the goal of preventing the policy from overfitting to the environment and relying on it to dictate behavior. Thus, without altering the core meaning of the tasks, we made the following modifications to create the OOD setting:

- **Seed and agent location** We change the seed and spawn location in the Minecraft world to perform the same task, and then the initial observation will not be identical to the training set.

- **Biome** We change the biome of the agent while keeping the task solvable. For example, change biome from plains to forest of task tool_pumpkin.

- **Tool** We modified the auxiliary tools while ensuring the tasks remained solvable. For example, in the survive_hunt, we replaced the iron_sword with diamond_axe.

- **Object location** We change the location of the object that the agent needs to interact with. For example, we changed the position of the stonecutter from being held in the hand to being placed in front of the agent.

For each task, we applied one or more of the aforementioned OOD modifications. It is important to note that the absolute performance in the OOD setting is not directly comparable to the baseline, as the tasks may become either easier or harder in the OOD environment.

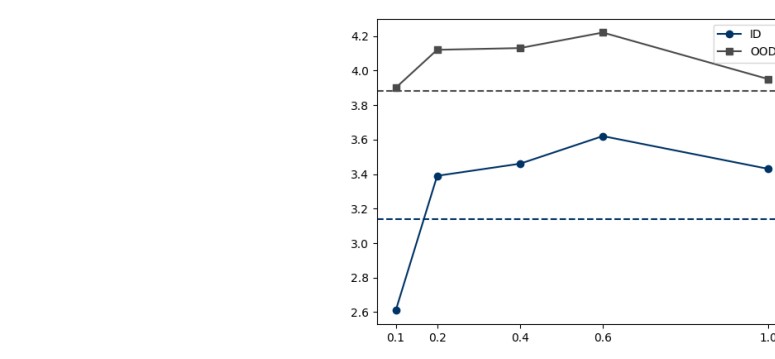

Figure 6: Different hyperparameter $\beta$ in task Collect Wood.

## B.5 HYPERPARAMETERS

Our training hyperparameters are listed in Table 6. We conducted a hyperparameter search on the "collect wood" task and used the same set of hyperparameters for all the other tasks. We visualized the performance of the "collect wood" task under different values of $\beta$. The result can be seen in Fig 6 The results showed similar performance when $\beta \geq 0.2$.

Table 6: Hyperparameters for training.

| Hyperparameter | Value |
|---|---|
| Optimizer | Adam |
| Learning Rate | 1e-2 |
| $\beta$ (in DPO) | 0.6 |
| Batch Size | Full Gradient Descent |
| Type of GPUs | NVIDIA RTX 4090, A40 |
| Training Precision | float32 |
| Data Collection Phase Samples | 500 |
| P-N Samples (each) | 150 |

## C  EXPERIMENT RESULTS

### C.1  BEHAVIOUR CLONING RESULTS

This baseline employs behavior cloning, trained exclusively on positive samples, without the inclusion of negative data or preference learning. We present results for both tuning soft prompt and the full parameters (Table 1).

### C.2  FULL FINE-TUNING RESULTS

We compare the results of our method with full fine-tuning. The latter involves ∼100M parameters, while the former only has 512 parameters, which is merely one in hundreds of thousands of the other. We found that in in-distribution settings, the soft prompt method achieves results comparable to those of full fine-tuning. However, in out-of-distribution (OOD) environments, soft prompt tuning outperformed across all tasks. The result can be found in Table 7.

### C.3  PARAMETER-EFFICIENT FINE-TUNING RESULTS

We conduct parameter-efficient fine-tuning on LoRA (Hu et al., 2021), BitFit (Zaken et al., 2022), VeRA (Kopiczko et al., 2024), and the result is in Table 8. In fact, all of these parameter counts are significantly larger than those of PGT. and the contrast is shown in Table 9.

Table 7: Comparisons between tuning soft prompt and full fine-tuning. The soft prompt method can bring better improvements than the counterpart, especially on OOD settings.

| Task | In Distribution (ID) | | | Out of Distribution (OOD) | | |
|---|---|---|---|---|---|---|
| | Pretrained | Full | Soft prompt | Pretrained | Full | Soft prompt |
| collect_wood | 3.14 | 3.46 | **3.62** | 3.88 | 4.04 | **4.22** |
| craft_stonecutter | 31.0 | **62.2** | 44.6 | 20.0 | 21.2 | **23.4** |
| explore_mine | 4.91 | 6.00 | **6.58** | 3.90 | 4.77 | **5.38** |
| tool_pumpkin | 48.3 | **58.4** | 57.8 | 16.6 | 22.2 | **25.8** |
| survive_hunt | 31.2 | **39.8** | **39.8** | 20.8 | 21.0 | **21.6** |
| obsidian | 42.0 | **62.2** | 57.2 | 4.2 | 6.0 | **8.2** |

Table 8: Parameter efficient fine-tuning result.

| Task | In Distribution(ID) | | | | Out of Distribution(OOD) | | | |
|---|---|---|---|---|---|---|---|---|
| | LoRA | BitFit | VeRA | PGT | Lora | BitFit | VeRA | PGT |
| collect_wood | 3.47 | 3.55 | 3.39 | **3.62** | 4.09 | 3.91 | 4.16 | **4.22** |
| craft_stonecutter | 49.4 | 48.6 | **52.2** | 44.6 | 19.8 | 18.0 | 18.8 | **23.4** |
| explore_mine | 6.52 | 5.37 | 5.76 | **6.58** | 5.17 | 4.42 | 4.67 | **5.38** |
| survive_hunt | 39.8 | 40.8 | **42.0** | 39.8 | 24.6 | 25.2 | **27.4** | 21.6 |
| tool_pumpkin | 50.4 | 56.2 | 52.8 | **57.8** | 19.6 | 20.8 | 22.4 | **25.8** |
| collect_obsidian | **71.2** | 55.8 | 57.8 | 57.2 | **10.6** | 6.2 | 2.6 | 8.2 |

## C.4 CONTINUAL LEARNING RESULTS

All of our continual learning baselines are based on fine-tuning the entire policy model, and the order of tasks for continual learning is as follows: collect_obsidian(⬛) → tool pumpkin(🎃) → craft_crafting_table(🔲) → explore climb(🧗). We implemented multi-task learning (MTL) (Table 3), naive continual learning (NCL) (Table 10), knowledge distillation (KD)(Table 11), experience replay (ER)(Table 12), and elastic weight consolidation (EWC)(Table 13).

## D OTHER PREFERENCE LEARNING ALGORITHMS

Our PGT method consists of data filtering and preference learning. The aforementioned experiments are all based on DPO for convenience, but other preference learning algorithms like IPO (Azar et al., 2024) and SLiC (Zhao et al., 2022; 2023b) are also possible. We experiment with IPO and SLiC [1] on the *goal latent representation* on several tasks and the results are listed in 14. It can be observed that both DPO and IPO improve task performance across different environments. Different tasks are suited to different algorithms (which may also be related to hyperparameters), but performance almost consistently improves after PGT, and a *goal latent representation* with just 512-dimensional parameters is sufficient.

---

[1]We choose **rank** calibration loss and **cross entropy** regularization loss, which is the same as SLiC-HF.

Table 9: The number of trainable parameters in full fine-tuning, PGT and other baselines.

|  | Full | LoRA | BitFit | VeRA | PGT |
|---|---|---|---|---|---|
| **# Parameters** | 86M | 393K | 80K | 15K | 512 |

Table 10: CL: naive continual learning. The task names in the first row represent the model trained up to the current task during sequential training (with both the pretrained model and PGT used as references); the task names in the first column represent the test results on each task. For brevity, craft_crafting_table is abbreviated as craft_table. To reduce human annotation costs, we do not test the results of explore_climb, but use it solely as a step in the training process. It is employed to examine the impact of later tasks on earlier ones during sequential training. The same principle applies to the subsequent tables on continual learning.

| Task | collect_obsidian | tool_pumpkin | craft_table | explore_climb | Pretrained | PGT |
|---|---|---|---|---|---|---|
| collect_obsidian | 6.0 | 4.6 | 7.0 | 6.8 | 4.2 | **8.2** |
| tool_pumpkin |  | 23.6 | 24.2 | 20.4 | 16.6 | **25.8** |
| craft_table |  |  | 5.2 | 7.0 | 6.0 | **18.4** |

Table 11: CL: knowledge distillation

| Task | collect_obsidian | tool_pumpkin | craft_table | explore_climb | Pretrained | PGT |
|---|---|---|---|---|---|---|
| collect_obsidian | 6.0 | 5.2 | 6.6 | 5.4 | 4.2 | **8.2** |
| tool_pumpkin |  | 24.6 | 23.4 | 20.6 | 16.6 | **25.8** |
| craft_table |  |  | 7.6 | 5.8 | 6.0 | **18.4** |

Table 12: CL: experience replay

| Task | collect_obsidian | tool_pumpkin | craft_table | explore_climb | Pretrained | PGT |
|---|---|---|---|---|---|---|
| collect_obsidian | 6.0 | 6.6 | 5.0 | 6.0 | 4.2 | **8.2** |
| tool_pumpkin |  | 22.8 | 21.8 | 25.0 | 16.6 | **25.8** |
| craft_table |  |  | 5.2 | 9.0 | 6.0 | **18.4** |

Table 13: CL: elastic weight consolidation

| Task | collect_obsidian | tool_pumpkin | craft_table | explore_climb | Pretrained | PGT |
|---|---|---|---|---|---|---|
| collect_obsidian | 6.0 | 8.2 | 5.4 | 5.4 | 4.2 | **8.2** |
| tool_pumpkin |  | 23.6 | 24.0 | 23.8 | 16.6 | **25.8** |
| craft_table |  |  | 5.0 | 7.4 | 6.0 | **18.4** |

Table 14: PGT with another preference learning algorithm - IPO and SLiC, on GROOT.

| Task | In Distribution(ID) | | | | Out of Distribution(OOD) | | | |
|---|---|---|---|---|---|---|---|---|
|  | Pretrained | DPO | IPO | SLiC | Pretrained | DPO | IPO | SLiC |
| collect_wood | 3.14 | **3.62** | 3.37 | 3.24 | 3.88 | **4.22** | 3.99 | 4.00 |
| craft_stonecutter | 31.0 | **44.6** | 42.0 | 37.0 | 20.0 | 23.4 | 23.0 | **25.6** |
| explore_mine | 4.91 | **6.58** | 5.44 | 6.34 | 3.90 | **5.38** | 4.70 | 5.29 |
| survive_hunt | 31.2 | 39.8 | 40.6 | **43.0** | 20.8 | 21.6 | **32.8** | 24.4 |
| tool_pumpkin | 48.3 | 57.8 | **62.2** | 60.6 | 16.6 | 25.8 | **30.6** | 27.6 |
| collect_obsidian | 42.0 | **57.2** | 50.4 | 34.4 | 4.2 | **8.2** | 4.8 | 3.4 |

