# OpenReview forum: "Optimizing Latent Goal by Learning from Trajectory Preference"
_ICLR.cc/2025/Conference — Submitted to ICLR 2025_

### Official Review · Reviewer_5JEp · 2024-11-01

**Soundness:** 2
**Presentation:** 2
**Contribution:** 2
**Rating:** 3
**Confidence:** 3

**Summary:**

The paper introduces PGT, Preference Goal Tuning which leverages the current policy to generate trajectories in simulation, using human preferences or already available simulation reward predictors to generate preferences, and then subsequently employs direct preference optimization (DPO) for downstream learning on the collected preference data.

**Strengths:**

The paper demonstrates strong results in Minecraft domain, when comparing against baselines.

**Weaknesses:**

The main weakness is the novelty of this approach. The approach uses online-DPO style method to generate and leverage preference data to learn downstream policy. Direct preference optimization approaches have already been applied to many domains. Although the improvements in results in Minecraft domain are appreciable, the paper does not offer any specific analysis for the pertaining application, which makes this work incremental.

Further, the method assumes trajectory generation for current policy in simulation, which might be cost-inefficient in real domains like robotics. This limits the applicability of the proposed approach in such scenarios.

The paper generates preference dataset using privileged information (preferences), however the comparison baseline Behavior cloning does not, which raises questions on the fairness of such comparison. It would have been interesting to see comparisons against other methods that leverage such preference data for policy learning like SLIC [1].

Therefore, I believe the paper is currently not ready for acceptance, and needs extensive updates to improve the novelty of the approach, writing and experiments for a stronger submission.

Minor comments:
1. Typo in Line 89
2. Fig 3 too small to read

[1] Calibrating Sequence likelihood Improves Conditional Language Generation

**Questions:**

None.

---

> ### Author Response · Authors · 2024-11-21
>
> Thank you for your dedication to reading our work and providing thoughtful feedback. Below are our responses to the weaknesses and questions you raised.
>
> # Weaknesses
>
> >The main weakness is the novelty of this approach. The approach uses online-DPO style method to generate and leverage preference data to learn downstream policy. Direct preference optimization approaches have already been applied to many domains. Although the improvements in results in Minecraft domain are appreciable, the paper does not offer any specific analysis for the pertaining application, which makes this work incremental.
>
> We believe that our approach represents a novel contribution as it is the first to propose using preference data for efficient post-training foundation policies. In fact, our approach is not limited to DPO; it can be applied to other preference learning methods. We add experiment of IPO[1] and SLIC. The results are presented in Appendix D. Additionally, our method requires a smaller amount of data and less training effort compared to traditional approaches, making it a resource-efficient solution. Furthermore, it offers broad applicability, such as serving as a continual learning method or integrating with higher-level planners. These capabilities underscore our contribution and demonstrate its potential impact across various open-world tasks and beyond.
>
> [1] Azar et al. A general theoretical paradigm to understand learning from human preferences. In International Conference on Artificial Intelligence and Statistics,pp. 4447–4455. PMLR, 2024.
>
> >Further, the method assumes trajectory generation for current policy in simulation, which might be cost-inefficient in real domains like robotics. This limits the applicability of the proposed approach in such scenarios.
>
> We believe that scenarios with high rollout costs is actually a favorable situation for PGT application. Unlike online RL and other imitation learning algorithms, which require a large amount of interaction, PGT requires very little data, typically around tens to a hundred trajectories. Therefore, the cost of data collection is relatively low. At the same time, the smaller data volume makes the process of annotating preferences less costly.
>
> >The paper generates preference dataset using privileged information (preferences), however the comparison baseline Behavior cloning does not, which raises questions on the fairness of such comparison. It would have been interesting to see comparisons against other methods that leverage such preference data for policy learning like SLIC [1].
>
> In fact, for the sake of fairness, when we used $N$ positive samples and $N$ negative samples to train PGT, we used $2N$ positive samples to train BC. Moreover, we have added experiments with SLIC as well. The results are included in Appendix D of the rebuttal version.

---

> > ### Comment · Reviewer_5JEp · 2024-11-25
> >
> > I do not agree with the author's comment that " it is the first to propose using preference data for efficient post-training foundation policies." since direct preference optimization approaches employ pre-trained (SFT) policies as reference policies and perform preference optimization while staying close to reference policy's distribution. If this is what the author's call "foundation policy", then I believe that this claim is wrong.
> >
> > Further, while I appreciate the additional experiments (SLIC and IPO), I am not convinced that the method offers sufficient novelty. The method is basically an online DPO style approach which includes additional data collection step before generating preferences. Although the authors call their method PGT, Equation 2 is just a token-level implementation of DPO [1]. Moreover, the authors derive their method in Appendix A by mentioning: "Following the derivation approach of DPO", but the derivation steps are almost exactly identical to DPO derivation. Therefore in its current form, I am not convinced that the paper adds sufficient research value for it to be accepted at the conference.
> >
> > [1] Rafael Rafailov el al: From r to Q∗: Your Language Model is Secretly a Q-Function

---

> > > ### Author Response · Authors · 2024-11-25
> > >
> > > Thanks for your review. First, PGT is a __general framework__ for preference learning, which can be applied to all kinds of preference learning, and not restricted to DPO. Although the derivation of the PGT loss in the paper is the same as that of DPO, when applying other preference learning methods such as SLIC and IPO, the derivation and the final form of the loss are completely different.
> > >
> > > Second, PGT provides an __efficient way__ to fine-tune the goal conditioned policies using preference data, requiring significantly less data and computational cost compared to traditional preference learning methods. PGT requires very little data, typically around tens to a hundred trajectories, making it highly suitable for use in environments where simulation is costly or in real-world scenarios.
> > >
> > > Furthermore, it offers __broad applicability__, such as serving as a continual learning method or integrating with higher-level planners. These capabilities underscore our contribution and demonstrate its potential impact across various open-world tasks and beyond.

---

> > > ### Author Response · Authors · 2024-12-02
> > >
> > > We’ll elaborate further. Our method is a __general approach to fine-tuning goal-conditioned policies using preferences__, which is why we named it __Policy Goal Tuning (PGT)__. Although in the paper we derived the method using DPO (Direct Preference Optimization), the PGT pipeline is actually applicable to __all preference learning methods__, not just limited to DPO. We have also added experimental results for SLIC and IPO. When using these two preference learning methods, the PGT loss is dependent on the specific method used. We hope this clarifies your confusion about our method being identical to DPO.
> > >
> > > At the same time, our method is an __efficient way to unlock the full potential of goal-conditioned policies__.  For a goal-conditioned foundation policy trained through supervised learning with large datasets, the latent goal space usually holds abundant semantic meanings. However, the instructions selected by humans might not map well to the optimal latent representation in the goal space. Our method aims to __obtain the optimal representation in goal space through a very small amount of training__. We believe this is novel.
> > >
> > > Experiments show that there are tasks that cannot be completed under ordinary human instructions, but through PGT fine-tuning, we have found several examples of successful outcomes. This demonstrates that, with minimal data fine-tuning, we have unlocked the capabilities of the goal-conditioned policy that were __inherently present but difficult to exhibit under human instructions__.
> > >
> > > We hope this elaboration can resolve your concerns. If you have further problems, please let us know.

---

### Official Review · Reviewer_m9MA · 2024-11-02

**Soundness:** 3
**Presentation:** 2
**Contribution:** 2
**Rating:** 5
**Confidence:** 3

**Summary:**

The authors present a framework to fine-tune latent goal via preference optimization for goal-conditioned policy. Especially, the proposed method yields an evident performance improvement with minimal fine-tuning of goal latent representation, and it presents potential capability in continual learning.

**Strengths:**

The proposed method is simple but effective in the presented applications and tasks.

It also presents potential for continual learning without fine-tuning policy model.

The authors conducted extensive experiments to illustrate the advantageous of the proposed methods in various tasks in Minecraft envrionement.

**Weaknesses:**

1) Limited application: The paper conducted experiments solely on Minecraft Skill forage, leaving questions how it works in other goal-conditioned applications.

2) This method can be hard to adopt in tasks where rolling out trajectories or obtaining preferences for them is costly

3) Utilizing negative trajectory samples seems critical in this method but it is already well known method. Unless there is a specific challenge to adopt such method in openworld task such as Minecraft, the contribution of this paper becomes marginal as it is just adopting good method in another applications.

**Questions:**

1) Can this method be easily adapted for other goal-conditioned tasks where the goal is not limited to a language prompt but could be, for example, an image—as in robotics tasks (as the authors mentioned), possibly Franka Kitchen [1,3] or UR3 BlockPush [2,3]? If not, an explanation of the limitations would help clarify the potential of the proposed method.

2) It is unclear to me how PGT works. Could you elaborate more on PGT and how to generate g in the first place to compute $L_{PGT} (g, g_{ref})$ ?

3) For each task, should we conduct iterative process which gathers trajectories and obtains preference them from human or reward models to get the optimal latent goal. If so, this method can be limited in generalization where the strength of the foundation model primarily lies.

4) In Table 2, is there any specific or presumed reason for STE+ not working well compared to STE in hunt task?

5) How sensitive is PGT with respect to  $\beta$ in Eq. 2? which value is used for each task?

6) Regarding weakness 3), could you explain how the proposed method differs from or improves upon existing applications of preference learning in open-world tasks?

Minor issue:
The same paper is referenced differently for Tajwar et al.

[1] Gupta, Abhishek, et al. "Relay policy learning: Solving long-horizon tasks via imitation and reinforcement learning." arXiv preprint arXiv:1910.11956 (2019).

[2] Kim, Jigang, et al. "Automating reinforcement learning with example-based resets." IEEE Robotics and Automation Letters 7.3 (2022): 6606-6613.

[3] Lee, Seungjae, et al. "Behavior generation with latent actions." arXiv preprint arXiv:2403.03181 (2024).

---

> ### Author Response · Authors · 2024-11-21
>
> Thank you for your dedication to reading our work thoroughly and providing thoughtful feedback. Your insights have been instrumental in helping us refine and improve our research. Below are our responses to the weaknesses and questions you raised.
>
> # Weaknesses
>
> >Limited application: The paper conducted experiments solely on Minecraft Skill forage, leaving questions how it works in other goal-conditioned applications.
>
> We believe that for many goal-conditioned policies, not limited to the open-ended world agent and robotics fields, PGT has potential applications. Due to constraints in time, computational resources, and model availability, we conducted experiments using two open-source foundation policies in the Minecraft domain. We consider other applications as future work.
>
> >This method can be hard to adopt in tasks where rolling out trajectories or obtaining preferences for them is costly
>
> We understand your question. We believe that scenarios with high rollout costs are actually __a favorable situation__ for PGT application. Unlike online RL and other imitation learning algorithms, which require a large amount of interaction, PGT requires __very little data, typically around tens to a hundred trajectories.__ Therefore, the cost of data collection is relatively low. At the same time, the smaller data volume makes the process of annotating preferences less costly.
>
> >Utilizing negative trajectory samples seems critical in this method but it is already well known method. Unless there is a specific challenge to adopt such method in openworld task such as Minecraft, the contribution of this paper becomes marginal as it is just adopting good method in another applications.
>
> We believe that our approach represents a novel contribution as it is the first to propose using preference data for __efficient post-training foundation policies__. This distinguishes our work from existing methods, even if they also leverage negative samples. Our method is applicable to nearly all goal-conditioned policies and is not limited to open-ended world environments or Minecraft. Additionally, our method requires __a smaller amount of data and less training effort compared to traditional approaches__, making it a resource-efficient solution. Furthermore, it offers __broad applicability__, such as serving as a continual learning method or integrating with higher-level planners. These capabilities underscore our contribution and demonstrate its potential impact across various open-world tasks and beyond.
>
> # Question
> Due to the word limit, we have included the responses to the questions in the next block.

---

> ### Author Response · Authors · 2024-11-21
>
> # Questions
>
> >Can this method be easily adapted for other goal-conditioned tasks where the goal is not limited to a language prompt but could be, for example, an image—as in robotics tasks (as the authors mentioned), possibly Franka Kitchen [1,3] or UR3 BlockPush [2,3]? If not, an explanation of the limitations would help clarify the potential of the proposed method.
>
> Thank you for your feedback. PGT can certainly be applied to other goal-conditioned tasks where the goal is not in natural language. In fact, Groot is a video-instruct foundation policy. We appreciate the two environments you provided, and we believe __PGT is well-suited for these scenarios__. We have added references to these papers.
>
> We take [3] as a concrete example. [3] employs VQ-BeT to construct a goal-conditioned foundation policy where the goal is provided as a sequence of images. After reading the code from [3], we found that these images are mapped into a _goal_seq_ via a _resnet_header. This is a latent representation of the goal. This is where PGT comes into play. The _goal_seq_ serves the same function as the _goal latent representation_ in our paper. We collect trajectories and label them as positive and negative samples, then train the goal_seq with PGT-loss. Suppose we obtain _goal_seq'_ after training. _goal_seq'_ can be used as a good latent to perform the task.
>
> [3]:Lee, Seungjae, et al. "Behavior generation with latent actions." arXiv preprint arXiv:2403.03181 (2024).
>
> >It is unclear to me how PGT works. Could you elaborate more on PGT and how to generate g in the first place to compute LPGT(g,gref) ?
>
> Take the GROOT policy and the "collect_wood" task as an example. GROOT follows an encoder-decoder architecture, taking video instructions as prompts. It encodes the video instructions with an encoder into a 512-dimensional latent vector, and the decoder uses this latent vector as a condition, acting as an auto-regressive policy network.
>
> During the data collection phase, we first select a video about collect wood, which may be suboptimal for GROOT. We can __encode the video using the encoder and obtain the latent goal, denoted as g_ref__. We then collect trajectories of collecting wood generate by the decoder and g_ref. Next, we label the trajectories as either positive or negative.
>
> In the training phase, we __initialize g as g_ref__ and use it as the condition for the decoder, then use previously collected preference data with PGT loss to train only the latent goal g. During this training process, the parameters of the decoder remain fixed.
>
> >For each task, should we conduct iterative process which gathers trajectories and obtains preference them from human or reward models to get the optimal latent goal. If so, this method can be limited in generalization where the strength of the foundation model primarily lies.
>
> Indeed, PGT trains a separate latent for each task, but PGT should be regarded as a plug-and-play module that generates a refined prompt, significantly enhancing the capabilities of goal-conditioned policies and even unlocking potential in the foundation model that ordinary prompting cannot. Since we only train the goal latent representation and do not alter the model parameters, PGT does not conflict with the generalization capabilities of the foundation model. When faced with unseen tasks, the foundation model can still function as a goal-conditioned policy seamlessly.
>
> >In Table 2, is there any specific or presumed reason for STE+ not working well compared to STE in hunt task?
>
> 1. To ensure consistency and convenience, we use the same hyperparameters for all tasks under the same loss and training parameter settings. We did not specifically tune hyperparameters for each task. Maybe the optimal hyperparameters differ from the default ones.
> 2. Actually STEVE-1 is not good at killing sheep. The quality of the collected synthetic data is not good enough.
>
> >How sensitive is PGT with respect to β in Eq. 2? which value is used for each task?
>
> We conducted a hyperparameter search on the "collect wood" task and used the same set of hyperparameters for all the other tasks. We visualized the performance of the "collect wood" task under different values of $\beta$. The results showed similar performance when $\beta \geq 0.2$. We choose $\beta=0.6$ for the rest of tasks. __The line graph of different $\beta$ and other hyperparameters used are listed in Appendix B.5.__
>
> >Regarding weakness 3), could you explain how the proposed method differs from or improves upon existing applications of preference learning in open-world tasks?
>
> Thank you for raising this important point. We addressed weakness 3 in our previous response. We hope our earlier explanation clarifies these differences.

---

> > ### Author Response · Authors · 2024-11-25
> >
> > As the rebuttal deadline approaches, We greatly look forward to hearing back from you. If there are additional questions or points that require further elaboration, we welcome them and will do our best to respond promptly and effectively before the deadline.

---

> > ### Comment · Reviewer_m9MA · 2024-11-25
> >
> > Thank you for the authors' effort in addressing some concerns. However, I still have some concerns.
> >
> > limited application: Although the authors mentioned that the proposed method could be applicable to other domains, such as well-adopted goal-conditioned tasks mentioned above, concerns still remain without solid experimental results. While it demonstrates some merits in post-training for the Minecraft task, it is questionable whether PGT would yield the same benefits in other well-adopted domains.

---

### Official Review · Reviewer_BsA8 · 2024-11-03

**Soundness:** 3
**Presentation:** 3
**Contribution:** 3
**Rating:** 6
**Confidence:** 4

**Summary:**

This paper presents a post-training method for foundation policies called Preference Goal Tuning (PGT). The PGT utilizes the preference trajectories either annotated by human experts or culmulative reward to fine-tune the latent goal representation in the goal-conditioned policies. In the experiments, PGT demonstrates the capability of increasing the performance of goal-conditioned tasks by fine-tuning relavent goal representaion structure in foundation models. PGT demonstrates good robustness in handling OOD tasks and surpass other baselines in the domain of Continual Learning (CL).

**Strengths:**

- The idea of utilizing preference data to fine-tune the foundation models is clear and intuitive. The cost of accquring prefence data is low, according to the claims in the paper.
- Fine-tuning only a small amount of parameters is proved effective in other related work (e.g. LoRA and other parameter efficient method). It is good to see the comparison between PGT, full fine-tuning and other parameter efficient methods. Moreover, the choice of only optimizing latent goal represenation is a good and intuitive when the foundation model is required to solve goal-conditioned tasks.
- It is good to see PGT is robust in situations where 1) with different initial prompt 2) OOD tasks.

**Weaknesses:**

- Minecraft environments is a good start, while more challenging tasks would be greatly appreciated.
- This is minor. I understand it is not easy to organize all sub-sections under section 4, but the title of the section 4 is not very informative. Please re-consider the title of section 4.
- "Trajectory Preference" is serious. I briefly read the derivation of the PGT loss in section A.2. From my understanding, the $\pi(\tau;g)$ denotes the action given by the policy is conditioned on the goal $g$ and the trajectory $\tau$. Under the MDP setting as indicated in section 2.1, the action is only conditioned the current state (plus goal $g$ in goal-conditioned problem). The $\tau$ in the $\pi$ should be explicitly degenerated to $s$, if I understand it correctly. I am not saying the derivation is totally wrong. In fact, the PGT loss is derived from DPO which handles bandits problem solely and I understand you treat the $\tau$ as a single step. You may consider rewording a bit to indicate PGT is still optimizing under bandits framework yet benefits from collected preference trajectories.

**Questions:**

- See the last argument in weakness. I would be happy and open to further discussions.
- If I understand it correctly that you do treat each $\tau$ as a single step to optimize under DPO, it is still required to model the whole task as a MDP? I know the tasks in this work are MDP problems and I am asking wether the non-Markovian problems can be solved using PGT. Looks to me you do not even need to emphasize the MDP in section 2.1. As long as you have the $\pi(\tau, g)$and $r(\tau)$, the PGT is valid.
- When annotating the trajectories, how to judge the quality of the trajectory if their goal $g$ is different?
- This is minor, why the counter-part of the Full Fine-Tuning in table 1 is the *Soft* Prompt? What does *Soft* indicate here?
- It is possible to use foundation model itself (or any other foundation model) to judge the quality of the trajectory?

---

> ### Author Response · Authors · 2024-11-21
>
> We sincerely appreciate the time and effort you have dedicated to carefully reading our paper, and we sincerely thank you for your valuable feedback. Below are our responses to the weaknesses and questions you raised.
>
> # Weaknesses
>
> >Minecraft environments is a good start, while more challenging tasks would be greatly appreciated.
>
> We completely agree with your perspective. We believe PGT is well suitable for other goal-conditioned policies in domains like robotics.
>
> For example, another reviewer mentioned a robotics environment UR3 BlockPush[1], for which we have provided implementation details.
>
> [1] Lee, Seungjae, et al. "Behavior generation with latent actions." arXiv preprint arXiv:2403.03181 (2024).
>
>
> >The title of section 4 is not very informative. Please re-consider the title of section 4.
>
> Thanks for your feedback. We revised the title of section 4 as __Experiments__ and we hope this title will be more informative.
>
> > "Trajectory Preference" is serious. I briefly read the derivation of the PGT loss in section A.2. From my understanding, the π(τ;g) denotes the action given by the policy is conditioned on the goal g and the trajectory τ.
>
> We are sorry for the misunderstanding. Assume $\tau = (s_t,a_t) (0 \leq t \leq N-1)$ is a $N$-step trajectory, $\pi(\tau;g)$ denotes the probability of generating $\tau$ conditioned on $g$ . We revise our notation as $\pi(\tau|g)$, and it can be expanded as:
> $$\begin{equation}
>     \pi(\tau|g) = \prod_{i=0}^{N-1} \pi(a_i|s_i,g) p(s_{i+1}|s_t,a_t)
> \end{equation}$$
> Where $p(s_{i+1}|s_t,a_i)$ is transition dynamics, which are model-independent. Therefore, it does not exist in the final loss.
>
> The revised version of derivation is in Appendix A. Despite some changes in notation, the final result remains the same.
>
> # Questions
>
> >If I understand it correctly that you do treat each τ as a single step to optimize under DPO, it is still required to model the whole task as a MDP?
>
> We believe this is a common misunderstanding. MDP problems refer to settings where actions can be made based solely on the current state without requiring information from past states. When facing a non-MDP problem, the usual practice in RL/agent is to expand state variables to include more information, and regard it as an approximate MDP problem. In the last step of loss derivation, we decompose $\tau$ as $(s,a)$ pairs. This step leverages the MDP modeling, as it is this step that enables us to compute the loss.
>
> Here we propose a further explanation: If there is a non-MDP environment where deciding on an action requires not only the current observation but also past observations, the state can be expanded to $[o_t, o_{t-1}, \dots, o_{t-h+1}]$ where $h$ is the context length. Under this modeling approach, __PGT is effective and can be used in this non-MDP environment.__
>
> >When annotating the trajectories, how to judge the quality of the trajectory if their goal g is different?
>
> There is no need to compare trajectories with different goals. The PGT approach involves collecting several trajectories for a specific goal, then distinguishing these trajectories as positive and negative samples for training. When labelling, any two trajectories are relative to the same goal.
>
> >This is minor, why the counter-part of the Full Fine-Tuning in table 1 is the Soft Prompt? What does Soft indicate here?
>
> Sorry for the unclearity here. Our method focuses on goal-conditioned policies. These policies encode goal to a latent representation. During training, we fixed the entire policy and trained the goal latent representation only, which is a 512 dimension vector. This is similar to soft-prompt in the field of Large Language Models, so we adopt the phrase here.
>
> Table 1 tries to offer a comparison between BC-loss and PGT-loss, and we performed comparisons under both fine-tuning goal latent (__which we called soft prompt__) and full parameter fine-tuning scenarios. We found that, regardless of the setting or task, PGT-loss consistently outperformed BC-loss.
>
> Moreover, for the ablation results regarding the use of soft-prompt fine-tuning versus full parameter fine-tuning, please refer to Figure 3.
>
> >It is possible to use foundation model itself (or any other foundation model) to judge the quality of the trajectory?
>
> You are absolutely right. We propose a potential approach here: adding a judgment head within the policy to assess the quality of the current action or previous action. The cumulative score of every action in a trajectory would then be used as the overall score for that trajectory.
>
> Using extra models to judge the quality of trajectories is another promising approach. For example, Video-to-text models, also known as video language models, can tackle various tasks like video question answering and video captioning. These models can be used to score trajectories.

---

> > ### Comment · Reviewer_BsA8 · 2024-11-21
> >
> > Thanks for your comments and it addressed many of my concerns. I will update the score to reflect it and happy to see if it got accepted.  Yes, the $\pi(\tau | g)$ make much more sense and now I can understand how do you derive equation (21). The comments on non-MDP settings is meaningful and I would encourage you to include it the paper.

---

> > > ### Author Response · Authors · 2024-11-25
> > >
> > > Thank you again for your thoughtful response and for taking the time to review our work in detail. We sincerely appreciate you updating the score and your kind words about our contributions. Your encouragement and constructive feedback mean a lot to us. We are grateful for your support and hope the work meets the expectations of the community.

---

### Official Review · Reviewer_EbcT · 2024-11-04

**Soundness:** 4
**Presentation:** 3
**Contribution:** 3
**Rating:** 8
**Confidence:** 4

**Summary:**

The submission introduces preference goal tuning (PGT), which is a form of continual learning approach to finetune the a pretrained model to tackle that prompt better. PGT embeds an initial prompt to a latent representation, and over several rounds of trajectory collection + finetuning of the latent representation via a modification of DPO's objective. Overall the results of PGT are sound and indicate that PGT can  improve prompt latent representations to 1. more robust and 2. even enable the solving of tasks that humans could not prompt out of pretrained models.

**Strengths:**

- PGT demonstrates that it can finetune a latent representation of a prompt to be more robust, with similar results to full finetuning on in-distribution finetuning data and stronger results on out-of-distribution settings (where the agent in the Minecraft world spawns elsewhere, in a visually/structurally different biome/background etc.).
- PGT results show that it outperforms human prompt engineering. It is also great to see the authors address the limitation of this method requiring preference data to work which is generally achievable in simulation via success conditions but not for the real world and is thus not free lunch.
- The design of PGT and training setup is fairly efficient and low-cost as the decoder is frozen
- Really interesting experiment with the "use trident" task having no success when given a normal prompt about trying to use the trident, but when the prompt's latent representation is finetuned via PGT there is some success now. It does indeed suggest the insight that pretrained foundation models have a lot of skills embedded in their weights, but need a good finetuned prompt to "unlock" that ability and leverage it. However is this the only skill where this occurs? Are there others and what are the exact numbers of the improvement (original model getting 0 success but with PGT the new latent prompt gets some nonzero success/return).

**Weaknesses:**

- Section A.2 about PGT loss mentions the assumption of a "Bradley-Terry-Model-liked oracle reward function" existing. I don't quite understand what it means to be a BT-model-liked reward function. Is there an example for some of the environments? Say collect wood or craft object X? If this is a reward function in the typical sense of RL literature, how dense/optimal does this reward need to be? This may be potentially a strong limiting factor of this work
- Section A.2 what does it mean the task can be subjective or objective? This needs elaboration as I don't understand how it relates to the derivations.
- This is perhaps is more of a opinion coming from a RL+embodied AI/robotics background but I find the term "Task Environment Generalization" to be a little misleading. Upon my first read I thought the OOD settings were different tasks (although hindsight that cannot make sense since you finetune for a specific task) as often times the word "environment" is interchangeably used with "task" (by e.g. gymnasium/gym libraries and various robot learning benchmarks). Maybe a better phrase is "Intra-Task Generalization" or something, I am not sure however.
- Table 2's choice of colors for improvement numbers vs worsened numbers is a bit strange to me (maybe a cultural thing). I would normally expect green values to be = good, red = bad, but it appears flipped here.

**Questions:**

- What were the initial prompts like used in Figure 4? I am not particularly surprised some finetuning on newly collected preference data for the wood collecting task would do well if the initial prompts were already fairly good.
- Is it possible to map the finetuned prompt latent representation to human text? That would be interesting from an interpretability standpoint + may unveil qualitative/quantitative insights into which prompts work for a model and what tend to not work.


Happy to raise my score. I think this is a well written and thought out paper and would love to have it accepted. My primary reason for keeping a 6 at the moment is the first 2 points in the weaknesses section. My big concern is about how good of a reward function you might need. For the feature of being able to e.g. elicit hidden skills, if the reward function to help finetune the prompt to elicit hidden skills in the pretrained model is hard to write / needs to be fairly dense and optimal the approach of PGT might not be easily scalable.

---

> ### Author Response · Authors · 2024-11-21
>
> We sincerely thank you for your valuable feedback and thoughtful comments! We truly appreciate the time and effort you have dedicated to reviewing our submission. Below, we provide detailed responses to your questions.
>
> # Weaknesses
>
>
> > Section A.2 about PGT loss mentions the assumption of a "Bradley-Terry-Model-liked oracle reward function" existing. I don't quite understand what it means to be a BT-model-liked reward function.
> We apologize for the unclearity. We will explain in detail.
>
> Generally, we want to utilize human preference to finetune our policy. "Preference" is assumed to be a subjective choice from human but generated by an oracle reward function $r^*(\tau)$, which is inaccessible. $r^*(\tau)$ represents how well trajectory. $\tau$ performs given task. The better $\tau$  performs, the higher $r^*(\tau)$ is. "Bradley-Terry-Model-liked oracle reward function" in the article refers to this. It is worth mentioning that __this oracle reward function is used only as a derivation target and is ultimately eliminated__. Its role is to help us derive PGT loss.
>
> We refine our writing and change some notations of Appendix A to make the derivation clearer. You can see the rebuttal version for more details.
>
>
> >Section A.2 what does it mean the task can be subjective or objective? This needs elaboration as I don't understand how it relates to the derivations.
>
> Sorry for the misunderstanding here. __This sentence is irrelevant to the derivation below. Our derivation is base on $r^*$. So it is fine to just ignore it.__
>
> "Task can be subjective or objective" means on data collection phase, the classification of trajectories can be based on human preference or _reward from environment_. Here _reward from environment_ is the singal from environment, which aligns with RL literature. For example, in the task "collect wood", we can simply define the number of logs collected as reward. We use trajectories that collect more logs as positive trajectories, and trajectories that collect less as negative ones. We call this kind of task "objective".
>
> For tasks like "explore climb", the agent needs to climb up a mountain, so no explicit reward can be obtained from the environment. Human labellers are required to annotate positive and negative trajectories. We call this kind of task "subjective".
>
>
> > This is perhaps is more of a opinion coming from a RL+embodied AI/robotics background but I find the term "Task Environment Generalization" to be a little misleading.
>
> We appreciate your perspective, and we understand "Out Of Distribution(OOD)" might initially seem potentially misleading. Our generalization refers to performing the same task in different locations, which is a crucial ability for open-ended world agents. For example, agents are required to collect wood in different biomes, which means they need to find different kinds of trees in different locations. Although we stick to the original term, we still admire your perspective.
>
> >Table 2's choice of colors for improvement numbers vs worsened numbers is a bit strange to me (maybe a cultural thing). I would normally expect green values to be = good, red = bad, but it appears flipped here.
>
> Thanks for this advice. We adopt it in the rebuttal version. It is truly a cultural thing :)
>
>
> # Questions
>
> >What were the initial prompts like used in Figure 4? I am not particularly surprised some finetuning on newly collected preference data for the wood collecting task would do well if the initial prompts were already fairly good.
>
> We collect 5 different initial reference videos by humans. They are normally collected wood videos in Minecraft without cherry-picking or special tricks. The results of iteration 0 are the performances of these initial prompts. The horizontal dashed line represents the best-performing initial prompt, and in the first small figure, the starting point of the iterations curve coincides with the horizontal line, indicating that this represents the best-performing initial prompt. The starting points of the iteration curves for other initial prompts are all below the horizontal line, and we want to emphasize that after iterations, they all rise well above the horizontal line, meaning that they all outperform the best initial reference video recorded by humans.
>
>
> >Is it possible to map the finetuned prompt latent representation to human text? That would be interesting from an interpretability standpoint + may unveil qualitative/quantitative insights into which prompts work for a model and what tends to not work.
>
> Although we have no way to convert the latent goal into text, that is a really interesting direction. In GROOT, the authors plotted the t-SNE of the latent representation. We print the t-SNE of our latent goal in different representations. Although we have not reached any conclusions, the latent goal space may have more properties worth exploring.

---

> > ### Author Response · Authors · 2024-11-25
> >
> > As the rebuttal deadline approaches, We greatly look forward to hearing back from you. If there are additional questions or points that require further elaboration, we welcome them and will do our best to respond promptly and effectively before the deadline.

---

> > > ### Comment · Reviewer_EbcT · 2024-11-29
> > > **Response**
> > >
> > > Thanks for the clarifications. I have no other concerns at the moment and have raised my score to an 8. It looks like solid work and it is nice that is fairly straightforward.

---

### Meta-Review · Area_Chair_zXLS · 2024-12-21

**Metareview:**

This paper proposes to finetune the goal representation from trajectory preference. The reviewers have mixed opinions. After the discussion, the main concerns on the applicability of this method beyond the minecraft environment demonstrated in the paper and the technical novelty this paper offers compared with many existing work based on online DPO. Although the authors added new experiments based on SLIC and IPO, they are still on the same minecraft domain. I think more experimental results on other domains are needed to support claiming the generality of the proposed method. Another issue raised in the review is the limited baselines. I suggest the authors to include other preference-based learning methods (to give some ideas, such as using the preference but to finetune different parts of the policy or with LoRA). The current comparison with BC is insufficient to establish the claims made.

Based on these considerations, I do not recommend accepting the paper as it is.

**Additional Comments On Reviewer Discussion:**

Reviewer EbcT raises questions on writing clarity and assumption. The concerns are addressed.
Reviewer BsA8 raises concerns on limited exp environments and writing clarity, The concerns are addressed.
Reviewer m9MA raises concerns on limited exp environment, limited applicability, and limited novelty. After the discussion, the concerns on applicability remains. While the algorithm applies, it's unclear how effective it is without experimental results.
Reviewer 5JEp raises concerns on the novelty and the choice of baselines (missing other preference-based methods). The author's claim " the first to propose using preference data for efficient post-training foundation policies." is not well supported, given the large body of works on LLM post-training using preference data. The reviewer still holds the opinion on the work having limited novelty compared with online DPO.

---

### Decision · Program_Chairs · 2025-01-22

Reject